# Development and Validation of Targeted Gene Sequencing Panel Based Companion Diagnostic for Korean Patients with Solid Tumors

**DOI:** 10.3390/cancers13205112

**Published:** 2021-10-12

**Authors:** Byung-Joo Min, Woo Seung Lee, Myung-Eui Seo, Kye-Hwa Lee, Seung-Yong Jeong, Ja-Lok Ku, Yeul Hong Kim, Sang-Won Shin, Ju Han Kim

**Affiliations:** 1DNA Analysis Division, National Forensic Service Seoul Institute, Seoul 08036, Korea; muz1978@snu.ac.kr; 2Seoul National University Biomedical Informatics (SNUBI), Division of Biomedical Informatics, Seoul National University College of Medicine, Seoul 03080, Korea; leefall2@snu.ac.kr (W.S.L.); iceblue31@snu.ac.kr (M.-E.S.); 3Asan Medical Center, University of Ulsan College of Medicine, Seoul 05505, Korea; geffa@amc.seoul.kr; 4Department of Surgery, Seoul National University College of Medicine, Seoul National University Hospital, Seoul 03080, Korea; syjeong@snu.ac.kr; 5Cancer Research Institute, Seoul National University, Seoul 03080, Korea; 6Korean Cell Line Bank, Laboratory of Cell Biology, Cancer Research Institute, Seoul National University College of Medicine, Seoul 03080, Korea; kujalok@snu.ac.kr; 7Department of Internal Medicine, Division of Oncology/Hematology, Korea University College of Medicine, Seoul 151-742, Korea; yhk0215@korea.ac.kr (Y.H.K.); shinsw@kumc.or.kr (S.-W.S.)

**Keywords:** precision medicine, targeted sequencing, custom panel, pharmacogenomics

## Abstract

**Simple Summary:**

We have developed and analytically validated the Korean Pan-cancer Companion Diagnostic (CDX) Panel to apply targeted anticancer drugs to Korean patients based on the molecular characteristics of tumors using tumor samples without matched patient normal samples. The panel included 31 genes with reported single nucleotide variants, 9 genes with reported copy number variations, and 15 genes with predictive responses to targeted drugs under clinical testing, enabling the panel to be analyzed for the targets of 30 targeted anticancer drugs. It is cost-effective and optimized for cancer type-specific therapy in Korean cancer patients across solid cancer types while minimizing the limitations of existing approaches. This gene screening method is expected to reduce test turnaround time and cost, making it a balanced approach to investigate solid cancer-related gene regions.

**Abstract:**

Recently, several panels using two representative targeting methods have been developed but they do not reflect racial specificity, especially for Asians. We have developed and analytically validated the Korean Pan-cancer Companion Diagnostic (CDX) Panel to apply targeted anticancer drugs to Korean patients based on the molecular characteristics of tumors using tumor samples without matched patient normal samples. The panel included 31 genes with reported single nucleotide variants, 9 genes with reported copy number variations, and 15 genes with predictive responses to targeted drugs under clinical testing, enabling the panel to be analyzed for the targets of 30 targeted anticancer drugs. It is cost-effective and optimized for cancer type-specific therapy in Korean cancer patients across solid cancer types while minimizing the limitations of existing approaches. In addition, the optimized filtering protocol for somatic variants from tumor-only samples enables researchers to use this panel without matched normal samples. To verify the panel, 241 frozen tumor tissues and 71 formalin-fixed paraffin-embedded (FFPE) samples from several institutes were registered. This gene screening method is expected to reduce test turnaround time and cost, making it a balanced approach to investigate solid cancer-related gene regions.

## 1. Introduction

Precision medicine based on genomic analysis is rapidly applied to the diagnosis and treatment of cancer. Next-generation sequencing (NGS) technology is specialized in large-scale genetic information analysis, making it a platform for the regular examination of tumors samples in clinical environments and the data is increasingly used in the diagnosis and treatment of patients. Compared to other sequencing platforms, NGS’s mass parallel sequencing capability provides a clear advantage by using a single limited input of nucleic acids to facilitate simultaneous screening of multiple markers from multiple samples for various genomic anomalies. For clinical applications, the target sequence of a limited set of clinically important genes was the most practical approach. In the case of solid tumors, many laboratories use relatively small pan-cancer panels to screen out about 50 (or fewer) genes, which maximizes sequencing capacity by investigating prominent mutant hotspots. This also helps to maintain reasonable costs and time required and to minimize the complexity of interpretation and reporting.

Recently several panels using two representative targeting methods have been developed and used as daily practices. Memorial Sloan Kettering-Integrated Mutation Profiling of Actionable Cancer Targets (MSK-IMPACT) is a hybridization capture-based panel for detecting not only protein coding mutations but also selected intron variants, copy number variants (CNV) and structural rearrangements in 341 (recently 468) cancer genes [1,2]. Paired tumor and matched normal samples are required to use this panel. FoundationOne CDx is another captured-based panel whose number of targeted cancer-related genes is 324 [3]. It provides information of small variants, CNVs, microsatellite instability (MSI) and tumor mutation burden (TMB). Those two panels were approved by the Food and Drug Administration (FDA) in the US [4]. Ion AmpliSeq Cancer Hotspot Panel (v2) and Illumina TruSeq Amplicon Cancer Panel are the representative amplicon-based commercial panels [5,6]. They were developed to analyze a selected small number of genes and genetic variants using small amounts of DNA, unlike the two panels mentioned first, and save time and experimental costs. Most of these currently developed and used panels are based on large-scale cancer genome studies, such as The Cancer Genome Atlas (TCGA) [7]. Therefore, genetic analysis based on the data from these panels is highly generalizable but does not reflect racial specificity. In particular, the number of Asian participants in the Cancer Genome Atlas (TCGA) is only 5.5 percent of the total. Since racial characteristics are very important in identifying the cause of the disease and treating patients, countries such as Japan are already working on developing ethnic-specific panels [8,9,10]. We focused on balancing the essential information acquisition, experimental costs, and turn-around time by considering the characteristics of previously developed and used targeted cancer panels [11,12]. We also aimed to increase clinical application utilization by including the genes associated with targeted therapy and actionable variants which are frequently observed in Korean patients [13,14]. 

In this study, we developed and analytically validated the Korean Pan-cancer Companion Diagnostic (CDX) panel to apply targeted anticancer drugs to Korean patients based on the molecular characteristics of tumors by using tumor samples without matched patient normal samples. For the design of the amplicons included in the panel, a list of previously reported genes related to solid cancer was compiled. Among the mutations occurring in these genes, genetic variants associated with targeted therapies currently in use or under clinical trials were prioritized to determine the extent of the genomic regions comprising the panel, which were subdivided into single nucleotide variants and copy number variations. The genetic variant list was broken down into groups of genes that could predict responses to commercially available targeted therapies, groups of genes that could predict responses to targeted drugs under clinical trials, and groups of genes that could predict responses or outcomes of targeted anticancer drugs in the future. Genetic variants that can predict the response to targeted therapies and help in prognosis were included in the panel for this study as a top priority. In addition, hotspots of the target genes of targeted anti-cancer drugs currently undergoing clinical trials were added to enable panels to maintain practicality and economic feasibility in the long term. The oligo primers were separated into groups that did not cause mutual interference as possible, enabling complex amplification by using multiple primers at the same time. The panel included 31 genes with reported single nucleotide variants, 9 genes with reported copy number variations, and 15 genes with predictive responses to targeted drugs under clinical testing, enabling the panel to be analyzed for the targets of 30 targeted anticancer drugs. To verify the panel, 241 frozen tumor tissues and 71 FFPE samples were registered. Based on the results of the validation, the analysis using the panel we developed in this study shows improvement from previous approaches. The panel includes both single-nucleotide variants and copy number variants of the genes involved in solid tumors, and the SNV variants of the individual samples identified by commercial hotspot panels and Sanger sequencing were all validated by the panel we developed. CNVs identified using array CGH were also validated. In addition, an optimized knowledge-based filtering method was applied to allow for the detection of the somatic variant with tumor-only data without germline variant information. 

The Korean Pan-cancer CDX Panel in this study will allow us determine how to treat patients and prescribe targeted anticancer drugs, by providing a profile of actionable and other driver alterations in the patients.

## 2. Materials and Methods

### 2.1. Panel Design and Target Region Coverage

The Korean Pan-cancer CDX Panel were designed for targeted sequencing of all exons and selected hotspot regions of 51 oncogenes based on their role in cancer targeted therapy. The panel includes 31 genes with reported single nucleotide variants and 9 genes with reported copy number variation (Table 1). In addition, 15 hotspots of genes that can predict responses to target drugs under clinical trials were added to allow for analysis of the targets of 30 targeted therapies including Vandetanib, Trastuzimab, Tofacitinib, Temibromiumus, Imatinib, Gefitinib, Everolimus, Erlotinib, Dabrafenib, Cruzotinib, Cobbitinib, Cetuximab, Ceritinib, Cabozozantinib, Axitinib, Affinityib, Avo-trazub, etc. For single nucleotide variants, oligo primers were designed to target the entire exome of the genes and thereby target all areas affecting the actual protein structure and expression. To detect amplification and deletion of the genes, three exons, including the first and last exon of the gene, were selected as the target. Oligo primer was produced to predict the copy number of the genes throughout the PCR tendency of the three exons, and the criteria for single nucleotide variants were applied for genes requiring analysis of both single nucleotide variants and copy number variation. The detailed target design is provided in the Appendix A.

### 2.2. Tumor Samples

The samples of tissues were collected from three institutes (Seoul National University Hospital, Korea University Anam Hospital and Asan Medical Center). DNA samples registered for validation were derived from frozen colon adenocarcinoma (COAD) 241 samples offered by Seoul National University Hospital, 61 FFPE COAD samples from Korea University Anam Hospital and 10 FFPE non-small cell lung cancer (NSCLC) from Asan Medical center. Biopsies were performed on diagnosed solid tumor patients; all of the patients were East Asian and their nationality was South Korean. Every registered patient signed a human-derived material research informed consent form provided by the Institutional Review Board. This study is approved by the Institutional Review Board at Seoul National University hospital. 

### 2.3. Library Preparation and Sequencing

DNA samples were extracted from frozen cancer tissues and FFPE tissue slides by using a QIAamp DNA Mini Kit and QIAamp DNA FFPE Tissue Kit (Qiagen, Hilden, Germany). Targeted sequencing was conducted using a custom AmpliSeq targeted panel to screen the genetic profile of the samples. The panel contained 1403 primer pairs that were multiplexed into two pools to avoid primer-dimer formation and interference during PCR. The range of amplicons amplified by these oligo primer pairs ranged from 125 to 175 bp, and the rate of ‘on target’ coverage for this panel was 98%. DNA fragment amplification and library construction were performed using the Ion Amplieq Library Kit 2.0 as described in the manufacturer’s instructions (Thermo Scientific, Waltham, MA, USA). The combined libraries were sequenced using the Ion Proton platform (Thermo Scientific, Waltham, MA, USA) and the sequencing quality statistics of the aligned BAM file were calculated using Picard CollectHSMetrics.

### 2.4. Single Nucleotide Variants and INDELs Calling

Genomic variants were called using the Torrent Mapping Alignment Program (TMAP), Torrent Suite Software V5.1 with customized parameter settings to identify somatic variants. The cutoff values for the variant calling parameters were default TMAP parameter settings for somatic variants except for two categories. Two cutoff values for allele frequency and coverage were changed as follows: allele frequency, >0.01 (SNV) and >0.02 (INDEL); coverage, >10 (SNV) and >20 (INDEL). Reads from the Ion Proton sequencer are heterogeneous in length, and additional filtering or trimming steps were not applied.

### 2.5. Sequencing Quality Control

The sequencing quality statistics including median depth, total aligned read, on-target coverage and percent coverage 10× of aligned BAM files were gained using Picard CollectHSMetrics [15]. In addition, uniformity of sequencing coverage with a percentage of base greater than 50% of the mean base coverage of the sample in the target region was used for comparison with hotspot panel.

### 2.6. Comparison Performance with Commercial Hotspot Panel

For the sequencing benchmark, the Ion AmpliSeq Cancer hotspot panel (V2), a commercial product for cancer sequencing, was used. For 27 genes shared by the Korean Pan- cancer CDX Panel and hotspot panel, we calculated uniformity based on depths of intersected regions of each panel’s target design files (bed file format). A total of 37 samples (20 FFPE samples, 17 frozen samples) were sequenced in both of the panels. The Wilcoxon signed rank test was used for uniformity of FFPE samples and frozen samples between the two panels. If the p-value was under 0.05, it was supposed to be statistically significant. We compared hot spot variants that were reported more than ten times by COSMIC (v88) in both panels [16].

### 2.7. Optimizing Somatic Variant Filtration

Because the aim of a cancer panel is assessing the actionable variant of tumor tissues without sequencing normal tissue, it would be necessary to distinguish somatic variants from germline variants for novel cancer variant research. Somatic tumor variant filtration strategies suggested in previous reports to optimize tumor-only molecular profiling using targeted next-generation sequencing panels were used [17]. The strategy for filtration is composed of four steps. 

First was the exclusion criteria that variants were retained only if they were reported in The Cancer Genome Atlas (TCGA) or the International Cancer Genome Consortium (ICGC) and thesomatic variant or the gene of variant is known to be an actionable gene in OncoKB or CIViC (Clinical Interpretation of Variants in Cancer) [18,19,20,21]. The second exclusion criteria was based on the population database and excluded variants if they were present at a minor allele frequency (MAF) more than 0.2% in any subpopulation in the 1000 Genome Project, Exome Aggregation Consortium (ExAC), Exome Sequencing Project (ESP), Korea Variant Archive (KOVA) or the Korean Genome Project [22,23,24,25,26]. To minimize the sequencing artifact, sequencing data from 18 normal blood samples that were not related to tested tumor samples were used for a germline pool dataset. The third exclusion criteria for the remaining variants got rid of variants presenting more than 0.2% allele frequency in the germline pool. The final step removed variants with “benign” or “likely benign” in Clinvar [27].

To evaluate filtering performance, 179 frozen tumor–blood normal paired samples were used. After the variant calling of each pair of samples, the hetero variant in a tumor sample that did not exist in the paired normal blood sample or homo alternative variant that was hetero variant in the paired normal blood sample were considered as a true somatic variant. However, the hetero variant in the tumor samples was supposed to be a false somatic variant if it was presenting as a hetero or homo variant in the paired blood normal sample.

### 2.8. Validation of the KRAS Variant with Direct Sequencing

To evaluate the reproducibility and specificity of the Korean Pan-cancer CDX Panel, we analyzed the consistency of previously confirmed genetic variants via Sanger sequencing for the 12th and 13th codons of the KRAS gene by using data of each individual variant call format (VCF) file of 62 FFPE samples. 

### 2.9. Copy Number Alteration Calling

For copy number detection without matched normal samples, Viscap was used. Viscap is a computational analysis tool used for inferring copy number alternations from targeted clinical sequencing data [28]. It calculates the ratio of sequence coverage to target intervals and computes the log2 ratio of each value to the median of the samples. Since the Korean Pan-cancer CDX Panel is an amplicon-based panel, input coverages were made from samples with uniformity (20%) greater than 80% by GATK DepthofCoverage without deduplication. The parameters of Viscap were that the threshold of the minimum exon was 6 and the threshold of the log2 cutoffs were −0.55 and 0.40, respectively.

To validate the copy number alteration which was inferred by Viscap, the orthogonal method, Agilent SurePrint G3 Human CGH Microarray 180K with Z-score algorithm (threshold 4) was applied to detect CNV.

### 2.10. Annotation

All variants from individual VCF files were annotated using ANNOVAR. The allele frequency of all subpopulations from the 1000 Genome project and Exome Aggregation Consortirum (ExAC), variant of ICGC and Clinvar (version 20190813) were downloaded from the ANNOVAR web database.

In addition, other sources, such as TCGA somatic MC3 data and the Korea Variant Archive (KOVA), were downloaded from their own web pages and processed for annotation [29]. OncoKB annotator was also used to classify oncogenic variants. If a variant was annotated ‘Splice site’, ‘Missense’, ’Nonsense’, ‘Inframe insertion’, ’Inframe deletion’, ’frame shift deletion’, ‘frame shift insertion’ or ‘Translation start site’, it was supposed to be a loss of function (LOF) variant. 

## 3. Results

### 3.1. Sequencing Performance

The average of total reads of frozen samples and FFPE samples were 1,669,840 (400,361–6,385,388) and 2,280,131 (636,130–5,499,590), respectively (Figure 1). The mean depth of sequencing was 1070.5× (±1292.25×) in frozen samples and 1278.9× (±2066.34×) in FFPE samples. The average covered region percentages of more than 10 depth on target region were 98% (sd: 2.1%) in frozen and 97% (sd: 2.6%) in FFPE. With these results, the sequencing shows sufficient depth to interrogate the target regions for somatic variants. There is no significant correlation between depth of coverage and GC contents (−0.018 with Pearson’s *p* < 2.2 × 10^−16^ in frozen samples and −0.012 with Pearson’s *p* = 0.72 in FFPE). Detailed statistics of sequencing quality have been included in Appendix A. 

Panel sequencing identified that the cohort (*n* = 312) had 26,389 SNV and 390 INDELs of all variants including synonymous germline variants. A total of 1,254 loss of function (LoF) oncogenic SNV and 64 LoF oncogenic INDELs were retained after filtering non-oncogenic variants (Appendix A). The number of missense alterations and nonsense variants are 1,114 (84%) and 297 (15%), respectively. The most frequent mutated genes were TP53 (78%, 244/312), KRAS (47%, 147/312), PIK3CA (24%, 74/312), BRCA2 (14%, 44/312), CEBPA (14%, 43/312) and FLT3 (14%, 42/412). In TP53, the most prevalent mutated codons were R175 (8%, 27/312), R248 (8%, 24/312), R273 (6%, 19/312) The most frequently altered codon in KRAS was G12 (30%, 94/312), followed by G13 (11%, 36/312).

### 3.2. Comparison of Performance with Commercial Hotspot Panel

The existence of 137 hotspot variants was confirmed in genome analysis using both the Korean Pan-cancer CDX Panel and the Ion AmpliSeq Cancer Hotspot Panel. However, 681 hotspot variants were exclusively detected using the Korean Pan-cancer CDX Panel (Figure 2). According to the results of uniformity comparison with the Ion AmpliSeq Cancer Hotspot Panel, however, uniformity of FFPE samples between the two panels showed no statistical significance and uniformity of frozen samples show statistically significant better uniformity (Appendix A). This suggests that the skewness of FFPE samples coverage distribution is presenting not only in the Korean Pan-cancer CDX Panel but also in other panel while the uniformity among frozen samples was better. In addition, the CDX Panel detected not only the variants that were called from the data using the AmpliSeq Cancer Hotspot Panel but also additional hotspot variants in other regions with stable sequencing quality.

### 3.3. Optimizing Somatic Variant Filtration

A total of 16,781 SNVs and INDELs variants were called as somatic mutation candidates from 241 frozen samples using the Korean Pan-cancer CDX Panel. Before the harmonic filtered steps were applied to the variants, the single steps show the highest percentage of true somatic variant was population DB filtration. (74.93%, Appendix A). However, actionable DB inclusion criteria decreased the true positive somatic variant percentage. (9.86%) The true somatic alteration percentage of the final filtered variants through all steps was 93.26% (Figure 3). The histogram indicates that there is no change before or after the Clinvar filter.

### 3.4. Reproducibility with KRAS Variant

To evaluate the reproducibility of the Korean Pan-cancer CDX Panel, 62 FFPE samples that had been analyzed as the KRAS variant using direct sequencing were registered (Table 2). A total of 6 KRAS variants, including Gly12Ala (c.35G>C), Gly12Asp (c.35G>A), Gly12Cys (c.34G>T), Gly12Ser (c.34G>A), Gly12Val (c.35G>T) and Gly13Asp (c.38G>A) have been identified from the 24 samples. These results suggest that all of the variants that appeared in the Sanger sequencing were also validated in the cancer panel (sensitivity 100%, 24/24). The specificity of the Korean Pan-cancer CDX Panel is 89.5% (34/38) when obtained from testing samples that have been checked as wild type with Sanger sequencing (Appendix A). Therefore, the accuracy of the test is 93% (58/62) with a positive predictive value (PPV) of 85.7%. Although there was discordance between the panel and the Sanger sequencing, it could be possible that the mutation could be detected because the Korean Pan-cancer CDX Panel method provides read-level resolution while Sanger sequencing is only providing a ratio with which it is difficult to distinguish between mutation and wild type. This result shows that the panel can detect actionable variants in precision oncology and verifies that it can replace the direct sequencing method as the primary scanning method to identify alterations contained by the patients.

### 3.5. Copy Number Alteration Calling

The results of the copy number alteration with Viscap were only two copy number alterations each in two samples (Figure 4). The minimum log2 ratio of EGFR is 2.54. The median and max of the log2 ratio are 3.01 and 3.57, respectively. In MYC copy number gain, the minimum log2 is 3.96. The median log2 ratio is 4.29 and the max log2 ratio is 4.66. The validation results obtained from microarray data with two algorithms (ADM-2 and Z-score) also show copy number amplification in EGFR (ADM-2: 2.4, Z-score: 249.2) and MYC (ADM-2: 3.91 Z-score: 107.7). The experimental data suggested that the copy number detection of the cancer panel was trustable because it was in concordance with different platforms. 

## 4. Discussion

With the development of cancer genomics and targeted treatments, the discovery of genetic variants in certain types of cancer has begun to change into a major factor necessary to treat cancer. The ability to profile clinically actionable genetic variants from cancer samples presents a new vision of individualized oncology, allowing for effective cancer treatment based on genotypes only. Since the next generation sequencing platform facilitates simultaneous screening of numerous markers for multiple patients, regular screening of genetic markers through large-scale parallel sequencing using the NGS platform is suitable for the clinical diagnostic testing of tumors. Clinical application of the NGS platform requires a panel design considering the characteristics of markers, the number of markers, the range of covered genes, and the type of genetic variants. Commercial hotspot panels have great strengths in detecting previously well-known genetic markers, but they do not cover the markers for various cancer species, and they cover only small regions of specific genes. These limits make them unable to screen for many types of potential abnormalities that may occur in cancers. Approaches through whole exon sequencing or whole genome sequencing can solve these problems, but they have an adverse effect on test turnaround time due to the limited number of multiplexing samples and increased interpretation complexity for the variants. In addition, they also cause a rise in the cost of testing.

Therefore, we have developed the cost-effective CDX Panel optimized for cancer type-specific therapy in Korean cancer patients across solid cancer types while minimizing the limitations of existing approaches. The sequencing quality of the panel we developed is better than the commercial hotspot panel for frozen samples. The analyzed results obtained using the Korean Pan-cancer CDX Panel showed that all hotspot variants detected using the commercial hotspot panel were confirmed and additional hotspot variants not detected in the commercial hotspot panel were also uncovered. 

The high concordance of KRAS variant detection between the CDX Panel and Sanger sequencing shows the high sensitivity of this panel. In addition, the optimized filtering for somatic variants from tumor-only samples enables researchers to use this panel. After validation of filtered tumor alteration with paired sequencing, only tumor sequencing without normal samples can find somatic variants with published genome databases. It will reduce cost of sequencing for not only targeted therapy but also research.

The Korean Pan-cancer CDX Panel is also designed to cover most of the cancer genome CNV for targeted therapy. There are other methods for copy number detection such as fluorescence in situ hybridization (FISH) and array comparative genomic hybridization (CGH) or qPCR. However, the former two methods could detect only large size CNV (5–10 Mbp for FISH, 10–25 kbp for CGH) and the latter method is a high-resolution method that could detect CNV that are too small [30]. Amplicons of the Korean Pan-cancer CDX Panel were developed to discover small CNVs that the FISH and array CGH methods could not detect and the Multiplexing PCR method for the panel is more economical than qPCR for covering most of the cancer genome. All CNVs detected through the Viscap algorithm were validated by array CGH experiments, but the frequency of CNVs identified throughout the test samples was too low (0.8%, 2/241). According to COSMIC, EGFR and MYC, the copy number gain rate in large intestine cancer is 3.48% and 7.8%, respectively, which is higher than the frequency of CNVs identified in the validation set of this study. This suggests that the Viscap CNV algorithm is too conservative to detect sufficient CNV rates and the development of CNV detection algorithms specialized for the Korean Pan-cancer CDX Panel might be necessary. 

This study has several limitations. First, the RNA panel used to analyze fusion genes has not been validated. The RNA panel was designed to analyze fusion genes as part of the CDX Panel. However, no fusion genes were from the DNA samples registered for the validation experiment, even though the reads from DNA amplification were generated in the target region. We determined that the cause of this limitation is that the fusion genes in solid cancer patients are very low, with less than 1%. Therefore, RNA panel validation is considered possible using the sample verified for the presence of fusion genes by existing test methods. Second, the four step somatic tumor variant filtration strategy used to optimize tumor-only molecular profiling using the targeted NGS panel used in this study had limitations in distinguishing somatic variants from rare germline variants because they are based on the common germline variant database. In spite of these limitations, validation experiments for the Korean Pan-cancer Companion Diagnostic (CDX) Panel show high sequencing quality and variant detection sensitivity for clinical applications.

In summary, we have developed a targeted gene sequencing panel which contains genes that are reported to be related to the onset of solid carcinoma and genes that are correlated with targeted treatment through various reviews.

We detected actionable variants of therapeutic targets for each cancer type on 21 genes that are association with 47 drugs (Table 3) [31,32,33,34,35,36,37,38,39,40,41,42,43,44,45,46,47,48,49,50,51,52]. The hotspots of these genes and all exon regions were included in the panel to maximize the clinical utility of previously known sequencing markers and additional variants that could be potential targets. 

## 5. Conclusions

This gene screening method is expected to reduce test turnaround time and cost, making it a balanced approach to investigate solid cancer-related gene regions.

## Figures and Tables

**Figure 1 cancers-13-05112-f001:**
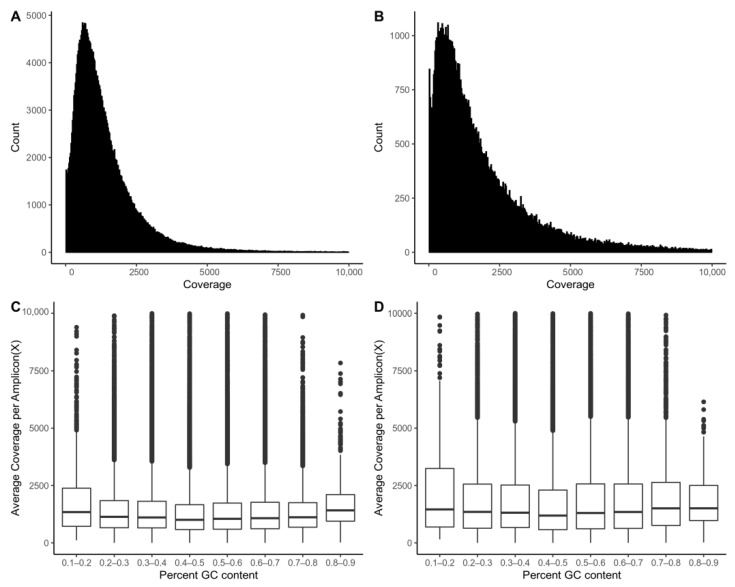
Uniformity of sequence coverage. (**A**) Distribution of sequence coverage of frozen samples (*n* = 241) across targeted amplicons by Korean Pan-cancer CDX Panel. (**B**) Distribution of sequence coverage of FFPE samples (*n* = 71) across targeted amplicons by Korean Pan-cancer CDX Panel. Each bar is 25 coverages. (**C**) Coverage for amplicons of frozen samples by percent GC content. (**D**) Coverage for amplicons of FFPE samples by percent GC content.

**Figure 2 cancers-13-05112-f002:**
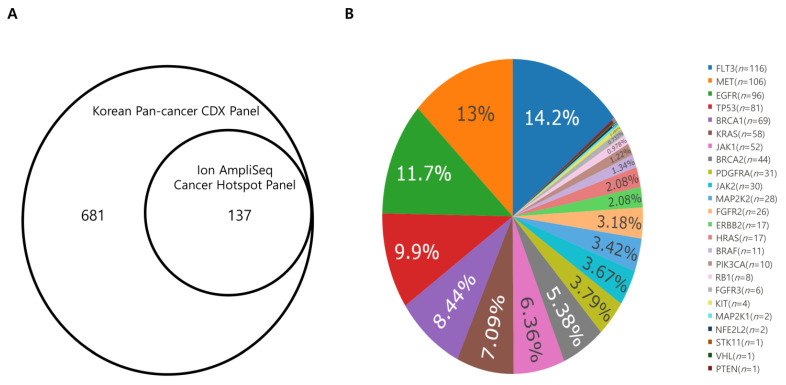
(**A**) Comparison of the hotspot panel in the Korean Pan-cancer CDX Panel and the Ion AmpliSeq Cancer Hotspot Panel over 37 samples. With Korean Pan-cancer Hotspot Panel, a total of 818 hotspot variants including all hotspot variants found using the AmpliSeq Cancer Hotspot Panel were detected (**B**) Pie chart of 818 SNVs in hotspot from 37 samples that were used to compare the CDX Panel and the Hotspot Panel. For example, a total of 116 SNV variants of FLT3 were found in 37 samples.

**Figure 3 cancers-13-05112-f003:**
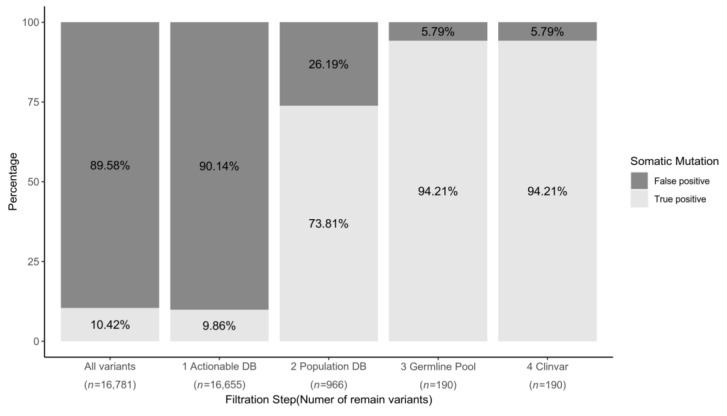
Bar graph of true somatic mutation (light gray) and false somatic mutation (dark gray) per each of the filtration steps. The filtration proceeds from the left bar to the right bar. Before the filtration process, 10.42% of variants were true positive somatic mutations. With the actionable gene database inclusion criteria, the false positive rate is increased even though variants are filtered. The true positive rate increase to 73.81% and 94.21% after population database and germline pool filtration, respectively. However, there is no filtered variant in the Clinvar-based step after three filtrations.

**Figure 4 cancers-13-05112-f004:**
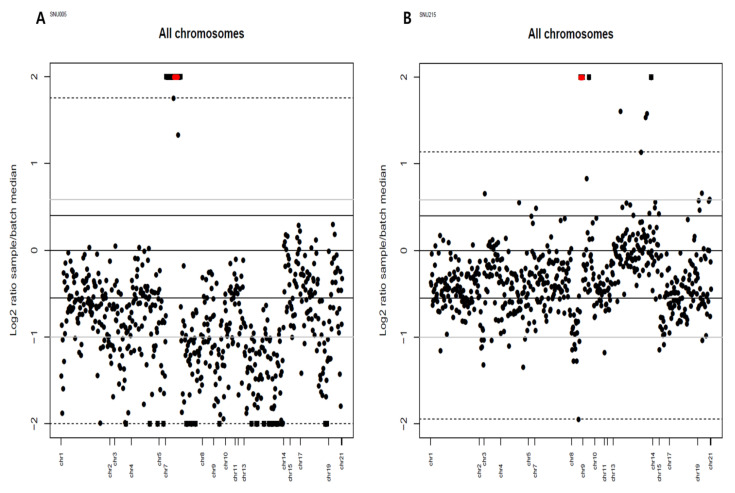
Copy number alteration plot of EGFR copy number gain (**A**) and MYC copy number gain (**B**). Each dark spot indicates a normalized sequencing read amplicon on a chromosome. The red dots are statistically significant copy number alterations using Viscap.

**Table 1 cancers-13-05112-t001:** Target gene lists in the CDX Panel. SNV—single nucleotide variants; CNV—copy number alteration.

Gene Name (Symbol)	Description	Target Alteration
*ERBB2*	Erb-B2 Receptor Tyrosine Kinase 2	SNV, CNV
*FGFR2*	Fibroblast Growth Factor Receptor 2	SNV, CNV
*FGFR3*	Fibroblast Growth Factor Receptor 3	SNV, CNV
*EGFR*	Epidermal Growth Factor Receptor	SNV, CNV
*MET*	MET Proto-Oncogene, Receptor Tyrosine Kinase	SNV, CNV
*MTOR*	Mechanistic Target of Rapamycin Kinase	SNV, CNV
*AKT1*	AKT Serine/Threonine Kinase 1	SNV
*BRAF*	B-Raf Proto-Oncogene, Serine/Threonine Kinase	SNV
*BRCA1*	BRCA1 DNA Repair Associated	SNV
*BRCA2*	BRCA2 DNA Repair Associated	SNV
*CEBPA*	CCAAT Enhancer Binding Protein Alpha	SNV
*CTNNB1*	Catenin Beta 1	SNV
*DDR2*	Discoidin Domain Receptor Tyrosine Kinase 2	SNV
*FLT3*	Fms Related Receptor Tyrosine Kinase 3	SNV
*HRAS*	HRas Proto-Oncogene, GTPase	SNV
*IDH1*	Isocitrate Dehydrogenase (NADP(+)) 1	SNV
*IDH2*	Isocitrate Dehydrogenase (NADP(+)) 2	SNV
*JAK1*	Janus Kinase 1	SNV
*JAK2*	Janus Kinase 2	SNV
*KIT*	KIT Proto-Oncogene, Receptor Tyrosine Kinase	SNV
*KRAS*	KRAS Proto-Oncogene, GTPase	SNV
*MAP2K1*	Mitogen-Activated Protein Kinase Kinase 1	SNV
*MAP2K2*	Mitogen-Activated Protein Kinase Kinase 2	SNV
*MYC*	MYC Proto-Oncogene, BHLH Transcription Factor	SNV
*NPM1*	Nucleophosmin 1	SNV
*NRAS*	NRAS Proto-Oncogene, GTPase	SNV
*PDGFRA*	Platelet Derived Growth Factor Receptor Alpha	SNV
*PIK3CA*	Phosphatidylinositol-4,5-Bisphosphate 3-Kinase Catalytic Subunit Alpha	SNV
*RB1*	RB Transcriptional Corepressor 1	SNV
*STK11*	Serine/Threonine Kinase 11	SNV
*TP53*	Tumor Protein P53	SNV
*CDK4*	Cyclin Dependent Kinase 4	CNV
*CDK6*	Cyclin Dependent Kinase 6	CNV
*ARID1A*	AT-Rich Interaction Domain 1A	Hotspot
*ATM*	ATM Serine/Threonine Kinase	Hotspot
*CDKN2A*	Cyclin Dependent Kinase Inhibitor 2A	Hotspot
*DNMT3A*	DNA Methyltransferase 3 Alpha	Hotspot
*FAT1*	FAT Atypical Cadherin 1	Hotspot
*FAT2*	FAT Atypical Cadherin 2	Hotspot
*KDM6AS*	Lysine Demethylase 6A	Hotspot
*MDM2*	MDM2 Proto-Oncogene	Hotspot
*NFE2L2*	Nuclear Factor, Erythroid 2 Like 2	Hotspot
*PTEN*	Phosphatase and Tensin Homolog	Hotspot
*RAC1*	Rac Family Small GTPase 1	Hotspot
*RHOA*	Ras Homolog Family Member A	Hotspot
*RUNX1*	RUNX Family Transcription Factor 1	Hotspot
*VHL*	Von Hippel-Lindau Tumor Suppressor	Hotspot

**Table 2 cancers-13-05112-t002:** Reproducibility of known KRAS Variant Samples.

KRAS Variant Status	Sample ID		Variant Statistics		Sample Statistics	
		Variant Calling	Alt Count *	Ref Count **	Mean Depth	On Target Cover Rate
Gly12Ala (c.35G>C)						
	FFPE 63	Positive	147	804	1206	0.99
Gly12Asp (c.35G>A)						
	FFPE 9	Positive	1113	3721	3604	0.99
	FFPE 14	Positive	230	1087	1150.733	0.99
	FFPE 18	Positive	246	510	832.0	0.99
	FFPE 29	Positive	91	353	538.9	0.99
	FFPE 39	Positive	24	295	594.7	0.99
	FFPE 55	Positive	862	1049	1171.9	0.99
	FFPE 56	Positive	428	2249	1682.4	0.99
	FFPE 85	Positive	213	1213	1352.0	0.99
	FFPE 90	Positive	291	699	1310.7	0.99
	FFPE 91	Positive	343	2618	2342.8	0.99
	FFPE 100	Positive	1156	4165	4310.8	0.99
Gly12Cys (c.34G>T)						
	FFPE 16	Positive	151	232	450.1	0.99
	FFPE 50	Positive	14	5	31.7	0.89
Gly12Ser (c.34G>A)						
	FFPE 118	Positive	547	2552	2505.1	0.99
Gly12Val (c.35G>T)						
	FFPE 5	Positive	134	793	1105.3	0.99
	FFPE 84	Positive	392	1091	1187.9	0.99
	FFPE 115	Positive	741	883	1623.3	0.99
	FFPE 116	Positive	296	570	1499.9	0.95
Gly13Asp (c.38G>A)						
	FFPE 15	Positive	115	1003	821.0	0.99
	FFPE 30	Positive	290	1954	1985.7	0.99
	FFPE 59	Positive	629	2582	2804.4	0.99
	FFPE 75	Positive	379	969	1485.9	0.99
	FFPE 104	Positive	1697	3056	3665.1	0.99

* Alt count—alternative allele read count; ** Ref count—reference allele read count.

**Table 3 cancers-13-05112-t003:** Actionable mutations were found in Korean Pan-cancer CDX Panel.

Gene	Drug	Number of Samples (*n* = 312)
*BRCA2*	Olaparib, Talazoparib, Rucaparib, Niraparib	200
*BRCA1*	Olaparib, Talazoparib, Rucaparib, Niraparib	173
*KRAS*	Trametinib, Cobimetinib, Binimetinib	164
	Cetuximab, Panitumumab	117
	AMG-510	5
*PIK3CA*	Fulvestrant, Alpelisib	100
*BRAF*	PLX8394	6
	Dabrafenib, Trametinib, Vemurafenib, Cobimetinib, Encorafenib, Binimetinib, Atezolizumab, Panitumumab	10
	Trametinib	2
*NRAS*	Cetuximab, Panitumumab	13
*PTEN*	GSK2636771, AZD8186	52
*HRAS*	Tipifarnib	11
*ERBB2*	Ado-Trastuzumab Emtansine, Neratinib	35
*AKT1*	AZD5363	7
*MTOR*	Everolimus, Temsirolimus	22
*ATM*	Olaparib	13
*PDGFRA*	Imatinib, Sunitinib, Regorafenib, Ripretinib	17
*FGFR2*	Debio1347, BGJ398, Erdafitinib, AZD4547	21
*MAP2K1*	Trametinib, Cobimetinib	17
*CDKN2A*	Palbociclib, Ribociclib, Abemaciclib	3
*KIT*	Imatinib	9
	Sunitinib, Regorafenib, Avapritinib, Ripretinib	2
*EGFR*	Afatinib	2
	Osimertinib	1
	Erlotinib, Gefitinib, Afatinib	2
	Poziotinib	1
	Erlotinib, Afatinib, Gefitinib, Osimertinib, Dacomitinib	1
*MET*	Capmatinib	2
	Crizotinib	2
*IDH1*	Ivosidenib	2
*FGFR3*	Debio1347, BGJ398, Erdafitinib, AZD4547	2

## Data Availability

The data presented in this study are available in Appendix A here.

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
