# Peer review of "Development and Validation of Targeted Gene Sequencing Panel Based Companion Diagnostic for Korean Patients with Solid Tumors"

_cancers, 2021, doi:10.3390/cancers13205112_

Round 1

Reviewer 1 Report

The authors  investigated their original way of detecting cancer-related-genes. It is quite reasonable concept that in-house domestic testing of genetic alterations in the expansion of sequencing technique era of these days. There are fundamental things.

1)the bias of both the sample size and the cancer-type

  241 of FF and 61 of FFPE colon cancer samples, and 10 FFPE lung cancer samples. Those are quite biased sample backgrounds. 

2)sample quality

   Of all the tissues, do the authors have any quality control rules? Ethical issues and informed consent status must be contained.

Author Response

Response to Reviewer 1 Comments

Point 1: the bias of both the sample size and the cancer-type

  241 of FF and 61 of FFPE colon cancer samples, and 10 FFPE lung cancer samples. Those are quite biased sample backgrounds. 

 Response 1: We agree with your comments. In the process of registering cancer-specific tissues, we did not obtain a sufficient number of lung scanner samples and some FFPE samples were excluded from the experimental validation process because they did not have the DNA field required for NGS experiments. We considered that experimental validation of the developed panel and analysis methods, the current research purposes, were not significantly affected by the biased sample backgrounds. However, we fully understand that the based sample backgrounds have a significant impact in analyzing the frequency and characteristics of mutations depending on the cancer type and we will register additional samples for future analysis.

Point 2: sample quality

   Of all the tissues, do the authors have any quality control rules? Ethical issues and informed consent status must be contained.

Response 2: Thank you for your valuable comments. We added a sentence in 2.2 Tumor Samples of Materials and Methods section as follow

Biopsies were performed on diagnosed solid tumor patients and all of the patients were East Asians and their nationality was South Korean. Every registered patients signed Human-Derived Material Research informed consent form provided by Institutional Review Board.

Reviewer 2 Report

My major recommendation to the authors to make the paper acceptable for publication is that they have to explain their study design, results and conclusions to a broader audience than their highly specialized field. If scientists working on cancer therapy or oncologists cannot get a clear understanding of the rationale, the methods and the conclusions, the study will not have been useful for the treatment of patients. Therefore, the authors need to explain all their specialized jargon for the benefit of other cancer researchers and oncologists.

Rather than merely listing the genes used in this study in the text, the authors should add a table indicating which gene the abbreviation refers to.

When describing a figure, please explain what the results mean rather than merely reiterating specialized language.

Figure 2, right panel, is difficult to see and decipher.

Please explain the sentence "Total..." line 291

Please explain Figures 3 and 4 in language that an oncologist or cancer scientist can understand.

What does "lacks coverage for the entire gene area" mean?

Please improve the grammar of the sentence starting on line 351.

All citations in brackets must be placed before the period at the end of the sentence.

The references should be corrected such that the titles are all lower case (unless indicating an abbreviation), except proper names and the first letter.

Please provide a reference indicating that the cancer panels are indeed different

Institution names (line 143) should be capitalized.

Minor but necessary corrections

In recent -> recently

Pan-caner -> Pan-cancer (throughout the manuscript)

of tumor -> of tumors

normal cell -> normal cell

the panel include -> the panel includes

The sentence staring on line 70 is too long and should be divided into two sentences.

trastuzumib -> trastuzimab

Line 151  DNA samples was were -> DNA samples were

10ng -> 10 ng

What are target bed files?

From previous report -> from previous reports

Please explain the sentence beginning with "Viscap"

INDELs were remained -> please explain

On the other hands -> Avoid using this term unless it is preceded by "On the one hand..."

...no statistical significant -> ...no statistical significance

Line 363  Change to: and the latter method is a high resolution method...

Line 388. genes those have -> please explain

Why is there an initial after "Cancer Genome..." in ref 7?
